# Arboviral Risk Associated with Solid Organ and Hematopoietic Stem Cell Grafts: The Prophylactic Answers Proposed by the French High Council of Public Health in a National Context

**DOI:** 10.3390/v15091783

**Published:** 2023-08-22

**Authors:** Bruno Pozzetto, Gilda Grard, Guillaume Durand, Marie-Claire Paty, Pierre Gallian, Sophie Lucas-Samuel, Stéphanie Diéterlé, Muriel Fromage, Marc Durand, Didier Lepelletier, Christian Chidiac, Bruno Hoen, Xavier Nicolas de Lamballerie

**Affiliations:** 1Haut Conseil de la Santé Publique, Ministère de la Santé et de la Prévention, 75007 Paris, France; marc.durand2@sante.gouv.fr (M.D.); didier.lepelletier@sante.gouv.fr (D.L.); christian.chidiac@univ-lyon1.fr (C.C.); bruno@hoen.pro (B.H.); 2GIMAP Team, CIRI-Centre International de Recherche en Infectiologie, Université Jean Monnet de Saint-Etienne, Université Claude Bernard Lyon 1, Inserm, U1111, CNRS, 42023 Saint-Etienne, France; 3Department of Infectious Agents and Hygiene, University Hospital of Saint-Etienne, 42055 Saint-Etienne, France; 4National Reference Center for Arboviruses, National Institute of Health and Medical Research (Inserm), 13005 Marseille, France; gilda.grard@inserm.fr (G.G.); guillaume.durand@inserm.fr (G.D.); xavier.de-lamballerie@univ-amu.fr (X.N.d.L.); 5French Armed Forces Biomedical Research Institute (IRBA), Valérie-André, 91220 Brétigny-sur-Orge, France; 6Santé Publique France, The French Public Health Agency, 94410 Saint-Maurice, France; marie-claire.paty@santepubliquefrance.fr; 7Etablissement Français du Sang, 93218 Saint-Denis, France; pierre.gallian@efs.sante.fr; 8Unité des Virus Émergents (UVE: Aix-Marseille Univ-IRD 190-Inserm 1207), 13385 Marseille, France; 9Agence de la Biomédecine, 93212 Saint-Denis, France; sophie.lucas-samuel@biomedecine.fr (S.L.-S.); stephanie.dieterle@biomedecine.fr (S.D.); 10Agence Nationale de Sécurité du Médicament et des Produits de Santé (ANSM), 93200 Saint-Denis, France; muriel.fromage@ansm.sante.fr; 11Department of Infectious and Tropical Diseases, University Hospital of Lyon, 69002 Lyon, France; 12Department of Infectious Diseases, University Hospital of Nancy, 54500 Vandoeuvre-lès-Nancy, France

**Keywords:** products of human origin, solid organ, hematopoietic stem cell, West Nile virus, dengue virus, tick-borne encephalitis virus, Usutu virus, chikungunya virus, Zika virus, France

## Abstract

Diseases caused by arboviruses are on the increase worldwide. In addition to arthropod bites, most arboviruses can be transmitted via accessory routes. Products of human origin (labile blood products, solid organs, hematopoietic stem cells, tissues) present a risk of contamination for the recipient if the donation is made when the donor is viremic. Mainland France and its overseas territories are exposed to a complex array of imported and endemic arboviruses, which differ according to their respective location. This narrative review describes the risks of acquiring certain arboviral diseases from human products, mainly solid organs and hematopoietic stem cells, in the French context. The main risks considered in this study are infections by West Nile virus, dengue virus, and tick-borne encephalitis virus. The ancillary risks represented by Usutu virus infection, chikungunya, and Zika are also addressed more briefly. For each disease, the guidelines issued by the French High Council of Public Health, which is responsible for mitigating the risks associated with products of human origin and for supporting public health policy decisions, are briefly outlined. This review highlights the need for a “One Health” approach and to standardize recommendations at the international level in areas with the same viral epidemiology.

## 1. Introduction

Arboviruses represent a vast group of viruses that are transmitted by arthropods, and include, for human diseases, hematophagous mosquitoes, phlebotomine sand flies, and ticks. Arboviral diseases are endemic to all continents except Antarctica. By definition, arthropod bites by breeding females constitute the major mode of transmission of arboviruses to human beings. However, other pathogen-specific modes of transmission have been identified, including milk or milk-derived foods from infected animals or women, transplacental transmission during pregnancy, sexual transmission from infected persons, or healthcare-associated transmission resulting from the administration of products originating from an infected donor (blood, organs, tissues, and cells). Table 1, partly inspired from a recent systematic review [1], shows the main arboviruses for which transmission via blood transfusion or grafts has been demonstrated or suspected.

This narrative review is specifically dedicated to the main risks generated by arboviral diseases potentially transmitted by solid organ and hematopoietic stem cell grafts in humans with special reference to the French context. As shown in Figure 1, France has a great geographical and complex administrative diversity, with 18 administrative regions divided, for most of them, into smaller areas named Departments (n = 101).

In terms of arboviral risk, the diversity in the geographical distribution of French territories leads to the circulation of a wide variety of vector species across space and time [9]. French territories are exposed to a variety of epidemiological situations, e.g., in mainland France, the absence of arbovirus circulation in certain areas, the circulation of endemic pathogens in animal cycles with occasional human spillovers (West Nile virus (WNV), Usutu virus (USUV), and tick-borne encephalitis virus (TBEV), and in recent years, in southern regions, the indigenous circulation of ‘tropical’ arboviruses such as dengue virus (DENV), chikungunya virus CHIKV), and Zika virus (ZIKV). In the overseas territories, dengue fever is endemic and epidemic in most tropical areas (Caribbean islands, French Guiana, islands in the south-west Indian Ocean, Pacific territories) and Zika and chikungunya have caused epidemics in all regions (except in the south-west Indian Ocean for Zika). Global warming is also a major factor affecting the distribution and vectorial capacity of arbovirus vectors [10], particularly in temperate zones that were previously protected from this risk.

Together with different national agencies such as (i) “Santé publique France” (SpF) (French public health agency) associated with different regional agencies for public health, (ii) the French Blood product agencies, namely, “Etablissement français du sang” (EFS) and “Centre de transfusion sanguine des armées” (CTSA), (iii) the “Agence de la biomedicine” (ABM) (French Agency of Biomedicine), the “Agence nationale de sécurité du médicament et des produits de santé” (ANSM) (National Agency for the Safety of Medicines and Health Products), and (iv) the “Centre national de reference (CNR) des arbovirus” (French national center for arboviruses), the “Haut conseil de la santé publique” (HCSP) (French High Council for Public Health) is in charge of defining recommendations in terms of the safety of products of human origin at the national level, including blood, solid organs, cells, and tissues. The aim of the current report is to focus on the arboviral diseases that are most frequently involved in solid organ transplantation (SOT) and hematopoietic stem cell transplantation (HSCT) in France. For each of these diseases, we will study the risks associated with SOT and HSCT with a rapid review of published cases of transmission at the international level, the epidemiological context in mainland and overseas France, and the specific prophylactic recommendations issued by the HCSP for recipients of solid organs and hematopoietic stem cells, in accordance with those of the European Center for Disease Prevention and Control (ECDC). Arboviral infections occurring in transplant recipients after mosquito bite, which accounts for most of the cases of arboviral infections in these patients, are excluded from the field of the review.

## 2. West Nile Virus (WNV)

WNV is a single-stranded RNA virus belonging to the *Flaviviridae* family. The main characteristics of WNV infection are shown in Table 1. The reservoir of the virus is constituted by wild birds that become infected mainly via mosquito bites (enzootic cycle). Infected mosquitoes (mostly *Culex*) can contaminate humans and other mammals, including horses; however, humans and horses are considered dead-end hosts, since the viral load in blood is not high enough to infect new mosquitoes [11]. Five distinct lineages have been identified, three of which have been linked to significant outbreaks in humans. First isolated in 1937 in Uganda from a febrile patient, WNV has been shown to circulate for several decades in Africa, Europe, the Middle East (and notably Israel), in certain parts of Asia, and even in Australia (lineage 1b, also known as Kunjin virus) [12]. Since the end of the 1990s, the circulation of WNV worsened in Europe, northern Africa, and the Middle East. In 1999, WNV was imported for the first time to the American continent, causing an outbreak of severe encephalitis in New York City [13]. Within 3 years, WNV disseminated throughout North America.

Of note, it was only after the virus became endemic in the USA that human-to-human transmission of WNV via contaminated blood or solid organs was recognized. The first donor-derived WNV infections were reported in 2002 [14,15,16]. Since then, about 40 patients have been shown to be contaminated via blood products, including 38 in the USA, mostly in 2002 before the implementation of nucleic acid testing (NAT) in blood donors [14,17], and 2 in Greece [18]. Among these, 29 recipients presented a symptomatic infection, including 26 neuroinvasive diseases and three febrile infections; two recipients remained asymptomatic, while no clinical data were available in the remaining nine cases. As for SOT-derived WNV infections, nine cases of transmission from deceased donors have been documented (Table 2), mostly in the USA [15,16,19,20,21,22,23] and in Italy [24,25]. As shown in Table 2, nine donors, who had been contaminated pre-mortem either by blood transfusion or by mosquito bite, were the source of these contaminations. Twenty-six subjects received organs from these nine donors: fourteen developed a neuroinvasive disease (including six fatal cases), one developed fever, nine stayed asymptomatic despite documented infection, and two were not infected. Interestingly, five donors were diagnosed by WNV PCR test in blood, while four exhibited only IgM-specific antibodies (Table 2), which demonstrates that both tests are effective for donors’ screening. No case of WNV transmission has been reported so far in HSCT recipients. Blood- and transplant-transmitted WNV infections significantly decreased after blood and organ donors’ screening, especially after the introduction of more sensitive NAT during the periods of WNV circulation.

In mainland France, WNV infections have been identified mainly in the southern part of the country. The first human cases were reported in 1962–1963 in the Camargue area and the first horse cases were only reported in the 2000s. The cases reported in humans, horses, and birds during the last 20 years in mainland France are shown in Figure 2. Three epidemic peaks were observed in 2003–2004, 2015, and 2018, respectively. During the 2022 season, six human cases and nine equine cases were observed (Figure 2). The low number of avian cases in France is probably due to the absence of systematic viral investigations in birds. Lineage 1a was predominant up to 2017, but lineage 2, originating from North Africa and Italy, is becoming endemic in the Mediterranean basin, possibly as a result of climate change [27].

In 2022 and 2023, the French HCSP updated its recommendations on the safety of blood-derived products, SOT, and HSCT about the risk of WNV transmission [28,29,30]. Its main recommendations are as follows:Human cases of WNV infection diagnosed on French territory must be informed as a notifiable disease to SpF [31];Regarding labile blood products, from the first confirmed autochthonous human case, an individual NAT screening (ID-NAT) must be implemented without delay for all blood donations collected in the affected Department; blood components prepared from donations already collected in the affected Department between the date of the alert and that of the effective implementation of the ID-NAT must be quarantined pending the results of the testing, with the exception of platelet concentrates that are all pathogen-reduced (Intercept Blood System^®^) and can be released without delay. Blood donors having stayed (at least one night) in an affected Department must be deferred until 28 days after their return or tested by ID-NAT. Candidates for donation reporting a diagnosis of WNV infection must be deferred 120 days after the end of symptoms;As for SOT and HSCT, in case of the identification of a new autochthonous case, donors living or having stayed in the Region or Department concerned must be tested by WNV NAT and IgM/IgG serology must be performed, ideally before transplantation. The decision of whether to use the organs or cells from donors testing positive either by NAT or IgM serology is submitted to the benefit–risk balance, with the recommendation of delaying the grafts that are not urgently needed;A list of countries at risk for WNV during the period of circulation of the virus (namely, June to November) is updated each year [32]. Candidates for blood, organ, or cell donations originating from or having traveled in these countries must be either tested for WNV (NAT for blood and NAT + serology for grafts) or their donation postponed for 28 days after returning from the risk area;In parallel, a surveillance of the circulation of WNV is recommended in horses and birds.

Until now, no case of WNV infection has been identified on French territory after blood transfusion, SOT, or HSCT. However, discussions are in progress for a more efficient organization of this surveillance, notably in collaboration with other European countries, in order to share epidemiological information in real time at the international level with blood and graft agencies and to improve the veterinary monitoring.

## 3. Dengue Virus

Dengue virus (DENV) is a single-stranded RNA virus belonging to the *Flaviviridae* family, which exhibits four serotypes numbered from 1 to 4. The main characteristics of DENV infection are shown in Table 1. Originally, the virus reservoir was wild non-human primates infected by mosquito bites in forests (sylvatic cycle). Infected mosquitoes of the *Stegomyia* subgenus of *Aedes* (mainly *Ae. aegypti* and *Ae. albopictus*) are capable of biting and infecting humans, but uninfected mosquitoes can also be infected while blood feeding in viremic humans. The resulting ‘human–mosquito–human’ urban cycle is extremely rapid and efficient, and is now responsible for the vast majority of human cases. The current global burden of dengue fever is considerable: 2.5 billion people in over 100 countries are exposed; 50 to 100 million DENV infections occur each year, with around 20,000 fatal cases [33]. Although frequently asymptomatic or presenting as a self-limited mild fever, dengue fever may be life-threatening and is mainly linked to capillary leak syndrome that can lead to shock and death, particularly in children from low-resource backgrounds. DENV circulates mainly in intertropical regions, but over the past 10 years, it has spread dramatically to temperate regions due to increased human and commercial exchanges, with a probable contribution from global warming, which has extended both the potential *Aedes* breeding areas and the mosquito activity period [34].

Despite the wide distribution of this “old” infection and the fact that the virus can be present for about one week in the blood of infected patients, the risk of dengue fever as a transfusion- or graft-transmitted disease emerged only recently [35]. The first documented case of post-transfusion dengue was observed in Hong Kong in 2002, but was only published in 2008 [36]. Between 2002 and 2019, a total of seven papers reported on 15 cases of the possible transmission of DENV to 15 recipients, with different levels of accountability [37,38,39,40,41,42,43]. All types of blood products were involved, with a majority of packed red cells. In terms of severity, six patients developed hemorrhagic dengue fever, six developed mild dengue fever, two remained asymptomatic, and the last case was not documented. In the two fatal cases, dengue fever was not deemed to be the cause of the death. Sabino et al. [41] showed that the risk of transmission was independent of the viral load measured in the blood product and much lower than that observed after mosquito bite. As for transplant-derived DENV infections, 10 cases of transmission from infected donors have been documented, including eight organ donors and two donors of bone marrow (Table 3) [43,44,45,46,47,48,49,50,51]; 13 recipients were involved, with a strong or very strong (same viral sequence between donor and receiver) accountability in 10 of them. The outcome was favorable in 10 of the 13 cases; at least one death could be attributed to the DENV infection (Table 3). A virological diagnosis was available in 7 of the 10 donors: 4 were positive for a direct test (NAT and/or NS1 antigen), 2 exhibited anti-DENV IgM antibodies, and 1 was positive for both direct (NAT) and indirect (IgM) markers. DENV transmission was also observed after.

In France, the epidemiological situation of dengue fever is particularly complex (Figure 1). In the French West Indies, the four serotypes of DENV have been circulating since the end of the 1990s and have caused several outbreaks [53,54,55] (Figure 3). In French Guiana, DENV serotypes 1–3 circulate actively. In these American French territories, *Ae. aegypti* is the main vector. On Indian Ocean Reunion island, a dengue fever epidemic broke out in 2018 (the last DENV outbreak was in 1977–1978). Initially, the main circulating serotype was DENV-2; however, since 2019, DENV-1 and DENV-3 serotypes have been increasingly more common, and in 2020, the most frequently isolated serotype was DENV-1 [56]. In the Indian ocean, *Ae. albopictus* is the most common vector. In French Polynesia, DENV is also endemic with co-circulation of the four serotypes; *Ae. aegypti* and *Ae. polynesiensis* are both responsible for the transmission of different arboviruses including DENV [57]. In mainland France, the first autochthonous cases of dengue fever occurred in 2010 [58]. From 2010 to 2021, 19 clusters totaling 48 subjects were recorded. In 2022, nine clusters of 65 subjects were identified in six different Departments. DENV-1 was identified in 21 cases and DENV-3 in 43 cases (first autochthonous detection of the latter serotype in mainland France) [59]. When identified, the index case was returning from an endemic area and was bitten by autochthonous *Ae. albopictus* mosquitoes that transmitted the virus in the vicinity. Currently, control strategies include systematic inquiries in the cluster areas, vector control operations, door-to-door information for local residents, and briefing of general practitioners for tracking mild cases. Despite these, it is more and more likely that the virus will become endemic in mosquitoes in relation to the extension of the vector system and its efficiency towards DENV, also favored by climate change [60,61].

The French HCSP recommended the following rules for the safety of blood products, SOT, and HSCT regarding the transmission of DENV:When a case or a cluster of dengue fever is identified in an endemic area, ID-NAT screening must be performed for blood donations at the Department level. When positive, the products must be discarded and the donor deferred 28 days from the date of the end of the symptoms. For donors of solid organs and bone marrow present in the cluster areas or having stayed there, PCR and IgM/IgG serology must be performed close to the time of the donation. In case of the positivity of one of these markers, it is recommended that living donors postpone the graft or select another donor if available; in deceased donors, it is recommended to discard the organs, except in case of vital emergency for the recipient for which a benefit–risk evaluation must be performed. If organs are transplanted, a specific follow-up of the recipient is required [62].In clusters of autochthonous dengue in mainland France, blood collections must be postponed in the area, donors living or having stayed in this area must be excluded from blood donation for 28 days, and blood products already collected in this area and not treated by the Intercept^®^ process must be placed in quarantine in order to be tested by ID-NAT. Each transmission event is considered closed at the end of 45 days following the onset of clinical symptoms of the last detected human case. In contrast to blood products, due to the very low risk of selecting a positive donor of solid organ or bone marrow, no specific information is provided in this context [63].

## 4. Tick-Borne Encephalitis Virus

Tick-borne encephalitis virus (TBEV) is a single-stranded RNA virus belonging to the *Flaviviridae* family; within the TBEV viral species, three main subtypes are defined by the International Committee on Taxonomy of Viruses: the European TBEV (TBEV-EU), the Siberian TBEV (TBEV-Sib), and the Far Eastern TBEV (TBEV-FE). The main characteristics of TBEV infection are shown in Table 1.

Ticks belonging to the *Ixodes ricinus* and *Ixodes persulcatus* species are considered the main natural vectors, but *Dermacentor reticulatus* was recently also shown to be an effective vector of TBEV [64]. The geographical distribution of TBEV is Europe and Asia (Figure 4) [65]. According to the stages of development of the tick, the natural reservoir of TBEV mainly constitutes small rodents, birds, and wild ungulates (mainly cervids) that are present in the natural environment of endemic areas (mainly forests and grassland areas with sufficient rainfall) [66]. Ticks can also acquire the virus transtadially (from larva to nymph to adult ticks), transovarially (from adult female tick to eggs), or when cofeeding on animals. Humans and domestic mammals, including dogs, goats, sheep, and cows, may be contaminated by tick bites. Besides tick bites, humans can be infected by the alimentary route via unpasteurized milk or milk products from animal origin (for reviews, see [67,68]).

Data on the transmission of TBEV by products of human origin are scarce [68]. In 1989, Wahlberg et al. [69] reported one case of the transmission of TBEV by blood transfusion to two recipients in Finland (year not mentioned, but between 1959 and 1987). The donor was sampled a few hours before the onset of clinical symptoms. The first recipient exhibited symptoms the day after transfusion, while the second one showed a typical biphasic infection with fever the day after transfusion and a neurological episode 14 days later. TBEV infection was confirmed in both patients. No further case of TBEV transmission via blood products was recorded. As for SOT, in 2017 in Poland, Lipowski et al. [70] reported three fatal cases of TBEV infection in recipients (liver and two kidneys) from a donor deceased following a car crash and living in a region endemic for TBEV. The symptoms appeared between 17 and 49 days post transplantation and presented as sepsis and meningitis in the liver recipient and encephalitis in the kidney recipients. All three patients died. In contrast to natural tick-borne encephalitis, the infection course was monophasic in the three cases and no abnormality was observed in the cerebrospinal fluid (CSF) for two of the three patients, which could have resulted from immune suppression. The diagnosis was performed retrospectively, based on next-generation sequencing that showed the presence of the TBEV genome in the brain or CSF of the three recipients.

Sequences of TBEV were also detected by RT-PCR in the brain (donor and two recipients) or CSF (one recipient), with similar sequences for the four patients. In mainland France, TBEV infection is mostly limited to three administrative regions of the east of the country (Grand-Est, Bourgogne-Franche-Comté, and Auvergne-Rhône-Alpes) (Figure 1 and Figure 3). In April 2020, an outbreak of encephalitis and meningoencephalitis occurred in the Ain Department (Auvergne-Rhône-Alpes region) where TBEV had never been detected before. Following a public health alert, 43 patients with encephalitis, meningoencephalitis, or flu-like symptoms were recorded; all of them but one had eaten fresh goat cheese made of raw milk originating from a single local producer. The alimentary transmission was evidenced by the presence of TBEV in a batch of cheese and goat milk [71]. Since June 2020, TBEV infection has become a notifiable disease in France [72]. From May 2021 to May 2023, 71 cases of TBEV infection were notified to SpF; 61 of them were autochthonous (including 17 in Departments where the virus had never been detected before), whereas 10 were acquired in other European countries. The latter observations are indicative of an increase in reported cases of TBEV infections in France over the last few years, as also observed across Europe [66], with many drivers that could explain this expansion including climate warming, modifications in human habits, and ecosystem changes [73].

The French HCSP recommended the following rules for the safety of blood products, SOT, and HSCT regarding the transmission of TBEV [74]:For blood products, donors having experienced a tick bite in the 28 days preceding the donation in an area known to be at risk for TBEV (in or out of France) during the period of virus circulation (March to November) must be excluded for 28 days after the tick bite’s date. In addition, blood collection is interrupted in areas where a source of foodborne outbreak of TBEV is recognized, with quarantine of blood products already collected and not secured by the Intercept^®^ process, until testing negative by TBEV NAT;For SOT and HSCT, living donors staying or traveling in at-risk areas for TBEV must be made aware of the risks of tick bites and of the consumption of unpasteurized milk and milk products from March to November to avoid contamination. All living donors should be questioned for a tick bite that had occurred less than one month before the donation in a zone at risk for TBEV when completing the pre-donation check-up list. In case of a positive answer to this question, living donors must be tested for TBEV (NAT and IgM/IgG serology) prior to the gift; if at least one of these tests is positive, it is recommended to postpone the graft or to select another donor if available. For deceased donors recently exposed to a tick bite, when this information can be recorded from his/her relatives or after skin inspection, as it may be difficult to obtain virological tests prior to the transplantation, it is recommended to inform the recipient(s) and their medical team of the potential risk of TBEV infection and to perform specific virological tests in case of fever or neurological symptoms in the two months following the transplantation.

## 5. Other Arboviruses Circulating in France

### 5.1. Usutu Virus

Usutu virus (USUV) is a single-stranded RNA virus belonging to the *Flaviviridae* family. This emerging arbovirus was first isolated in 1959 in Ndumu, Natal, South Africa. Restricted for a long time to sub-Saharan Africa, USUV was introduced to Europe in 1996 with approximately one hundred cases reported so far in this area [75]. As shown in Table 1, its characteristics are very close to that of WNV: similar reservoirs (wild birds), similar vectors, humans and other mammals (including horses and wild boars) as dead-end hosts, and similar clinical pictures. It was first isolated in southern France (Occitanie region) in birds in 2015 [76] and in humans in 2016 [77]. The second French case occurred in 2022 in the Nouvelle-Aquitaine region; because of former vaccination against yellow fever and dengue infection in this patient, the diagnosis of USUV infection was made possible only by seroneutralization assay. So far, USUV has not been shown to be transmitted via products of human origin. However, following its spread in many European countries via different lineages, attention should be paid to this risk in the future [30].

### 5.2. Chikungunya Virus

Chikungunya virus (CHIKV) is a single-stranded RNA virus belonging to the *Togaviridae* family and *Alphavirus* genus. As shown in Table 1, CHIKV is transmitted by *Aedes* mosquitoes and is responsible for a high percentage of symptomatic forms mainly represented by joint pains that can evolve to chronic rheumatism. First isolated in 1952 in Tanzania in a patient suffering of arthralgia (the name “chikungunya” derives from a word in the Kimakonde language meaning “to be bent with pain”), the virus was rapidly shown to circulate in other regions of Africa and Asia. Different lineages have been described according to their geographical origin: West African (WA), East, Central and South African (ECSA), and Asian lineages. In 2004, an “epidemic lineage” derived from ESCA strains and named Indian Ocean lineage (IOL) emerged in Kenya; due to mutations of envelope proteins that facilitate its adaptation to *Ae. albopictus*, this lineage disseminated to southeast Asia, the Indian subcontinent, territories of the Indian Ocean and Pacific Ocean, and Europe [78,79]. In parallel, strains derived from the original ESCA lineage and the Asian lineage disseminated to the American continent with *Ae. aegypti* as the main vector. CHIKV is genetically close to other viruses of the *Alphavirus* genus such as O’Nyong Nyong, Ross River, and Mayaro viruses.

As with DENV, the initial sylvatic cycle involved forest mosquitoes and non-human primates, but the ‘human–aedes–human’ urban cycle is now responsible for the majority of epidemic cases in humans. Human-to-human transmission of CHIKV is exceptional, with a few cases of infection in utero or during childbirth [80]. The virus has never been isolated from sperm or human milk. Infectious CHIKV was isolated from positive CHIKV RNA qRT-PCR corneoscleral rims from four potential corneal donors living in La Réunion during the 2005–2006 outbreak of CHIKV infection [7]. Despite the probable exposure of recipients to infected products, products of human origin have never been found to be responsible for human contamination.

In France, different outbreaks of CHIKV infections were recorded:The Island of La Réunion was hit by a huge outbreak in 2005–2006 with more than 300,000 cases (one third of the whole population) [81,82]. No significant re-emergence of the virus was further observed.The French Territories in the Americas also experienced an important CHIKV outbreak in 2013–2014 with respectively 72,500, 81,200, and 15,000 cases in Martinique, Guadeloupe, and French Guiana. No significant re-emergence of the virus was further observed.In French Polynesia, a CHIKV outbreak was reported in 2014–2015.Finally, three clusters of autochthonous CHIKV infections were observed in the south of mainland France: 2 cases in 2010 [83], 12 cases in 2014, and 17 cases in 2017. These emerging cases must be taken into consideration as the capacity of adaptation of this virus to its vector is illustrated by the size of the first outbreak that occurred in Europe in 2007 and that had involved 217 cases in eastern Italy [84,85].

Despite the absence of the demonstrated transmission of CHIKV by blood products, CHIKV NAT was implemented in the French overseas territories and also in the mainland departments experiencing cases of autochthonous infections. Persons returning from regions where the virus circulates are excluded from blood donation for 28 days. The same nonspecific measures as for dengue fever are recommended for CHIKV infection (see above). For donors of SOT and HSCT present or having stayed in these areas, NAT and IgM/IgG serology must be performed close to the time of the donation. In case of the positivity of one of these markers, it is recommended that living donors postpone the graft or select another donor if available. For HSCT, if it is not possible to postpone or find another donor, a risk-based approach and a specific follow-up of the recipient are required. In deceased donors, it is recommended that the organs be discarded, except in case of vital emergency in the recipient for whom a benefit–risk evaluation must be performed. If the organs are used, a specific follow-up of the recipient is required [86]. These guidelines are only founded on a precautionary principle, since it has been assessed that transplant recipients who developed CHIKV infection after mosquito bite did not present with severe infection [87,88].

### 5.3. Zika Virus

After the first isolation of the Zika virus (ZIKV) in a monkey in the Zika forest, Uganda, in 1947, and in *Ae. africanus* mosquito, which suggested its arboviral origin, the first case of human infection was reported in 1952 [89]. Again, the epidemiological cycle of Zika changed from its original sylvatic cycle to a predominantly ‘human–aedes–human’ urban cycle. As shown in Table 1, ZIKV is transmitted by various vectors, including *Ae. aegypti* and *Ae. albopictus*. ZIKV raised relatively little medical attention until the report of an outbreak in Micronesia in 2007 [90], and of a second one in French Polynesia in October 2013 [91], which allowed the usual mild clinical presentation of the disease when symptomatic to be described. The second outbreak also allowed the identification of an increased risk of perinatal transmission [92] and of Guillain–Barre syndrome [93,94]. Then, CHIKV reached New Caledonia in January 2014 and South America and the Caribbean in 2015–2016 [95,96]. The latter outbreak, which led to the issue of a Public Health Emergency of International Concern by the World Health Organization (WHO) in February 2016, was a significant shift in the epidemiology of Zika infection: (i) there was a huge extension to vast territories of the American continent; (ii) after a first report in 2011 [97], the sexual transmission of the virus via semen was confirmed [98]; (iii) the vertical transmission was shown to happen frequently and a description of severe disease in the newborn was reported, with a notably high proportion of microcephaly and other malformations [99,100,101]. ZIKV is a genetically versatile virus with two major lineages: the African lineage that was predominant up to the 1970s and that was mostly transmitted to humans via a sylvatic cycle, and the Asian lineage that adapted notably to *Ae. aegypti* and was responsible for large outbreaks via an urban cycle in southeast Asia, the Pacific Territories, and the American continent [102,103].

In terms of blood safety, ZIKV was shown to be able to persist in some cases for up to two months after the onset of symptoms in serum or plasma [104]. To date, four cases of transfusion-borne ZIKV infection have been reported [105,106,107], all of them in Brazil: three occurred after the transfusion of platelets and one after the transfusion of packed red cells. Of the three living transfusion recipients, none displayed any clinical sign of ZIKV infection. After the implementation of NAT for preventing the transfusion of blood products contaminated by circulating arboviruses (DENV, CHIKV, and ZIKV) in French Polynesia between 2012 and 2018, it was shown that this measure excluded 5 blood donations reactive for DENV RNA, 34 for CHIKV, and 42 for ZIKV. As Zika screening could not have been implemented before the third month of the outbreak, 36 blood products from ZIKV-infected donors were transfused to 26 recipients, which resulted in no transfusion-transmitted ZIKV infection [108]. Altogether, these data illustrate that if ZIKV could be transmitted very unfrequently via labile blood products, the clinical consequences remain limited. As for solid organs, bone marrow, and tissues, no case of ZIKV infection has been yet reported.

As described above, the outbreaks of Zika having occurred in the French overseas territories (French Polynesia in 2013, New Caledonia in 2014, and French territories in the Americas in 2015–2016) contributed largely to the description of the clinical and epidemiological characteristics of this emerging disease. Interestingly, the first episode of Zika in Europe was reported in southern France (Provence-Alpes-Côte d’Azur region) in 2019 [109,110]. The index patient exhibited fever, rash, and fatigue and tested positive for ZIKV by NAT in blood two days after the onset of symptoms. IgM antibodies were also positive together with ZIKV-specific neutralizing antibodies in a blood sample collected 12 days after symptom onset. The door-to-door campaign conducted in the vicinity of the index case led to the identification of two more healthy patients who had developed fever and rash 8 days before the onset of the rash in the index case and who lived within 90 m from the index case; both of them also tested positive for ZIKV IgM and IgG and were confirmed positive for ZIKV by seroneutralization. None of the three patients or their partners reported recent travel to a ZIKV-endemic area. No further case of autochthonous Zika was reported in mainland France during the following years.

Since February 2016, Zika has become a notifiable disease in France [111]. ZIKV NAT was implemented for blood donors in the overseas territories where the virus circulated widely (New Caledonia, French Polynesia, and French America Departments). Patients exhibiting ZIKV infection are excluded for 28 days after the onset of clinical symptoms or the date of biological diagnosis. Persons returning from an epidemic zone or having had sex with an infected partner (less than 6 months before donation if the partner is a man and 2 months if she is a woman) are excluded for 28 days from blood donation. During the outbreaks, the same nonspecific measures as for dengue fever are implemented for ZIKV infection (see above). Despite the absence of reported cases of ZIKV transmission in SOT or HSCT, similar specific measures to those recommended for CHIKV infection are recommended (see above). Following the autochthonous episode notified in mainland France in 2019, due to the limited size of the outbreak (three cases) without further development, the HCSP required no specific measure for either blood safety or organ and cell grafts [112].

## 6. Concluding Remarks

This review provides an overview of the main arboviral risks observed in mainland and overseas France, with a particular focus on the safety of SOT and HCST. With global climate change, the dramatic expansion of the range of certain competent vectors, the growing internationalization of exchanges of people and goods, and the increased demands in terms of the quality of products of human origin, the viral risks multiply, adding difficulty to the selection of donors of solid organs or hematopoietic stem cells. At the same time, the need for products of human origin is increasing as a result of advances in medical care for many chronic diseases, and the refinement of donor selection criteria, particularly from a virological point of view, may limit the availability of organs and cells, requiring a careful assessment of the benefit–risk ratio in each individual situation.

This overview illustrates some of the diversity of arboviral risk in France despite the relatively small area of the country at the world scale. The role of the French HCSP, in collaboration with other national agencies and the French reference center for arboviruses, is to define clear, pragmatic selection criteria to propose adequate screening of donors and to help clinicians to make informed choices with regard to the selection of donors and the follow-up of recipients. The criteria need to be constantly updated and specific to a given region, as outbreaks occur in different geographical contexts.

After each new viral episode, the current recommendations need to be checked to ensure they are still relevant and to be interrupted when the circulation of a viral agent ceases. As arboviral diseases are mainly zoonoses, a “One Health” approach is imperative. Finally, international cooperation (WHO and ECDC, in particular) is desirable in order to standardize recommendations between countries, as viruses, birds, and even insects know no borders, and the diversity of recommendations can be detrimental to exchanges of products of human origin between foreign partners involved in transplantation. We hope that this review will contribute to this joint effort.

## Figures and Tables

**Figure 1 viruses-15-01783-f001:**
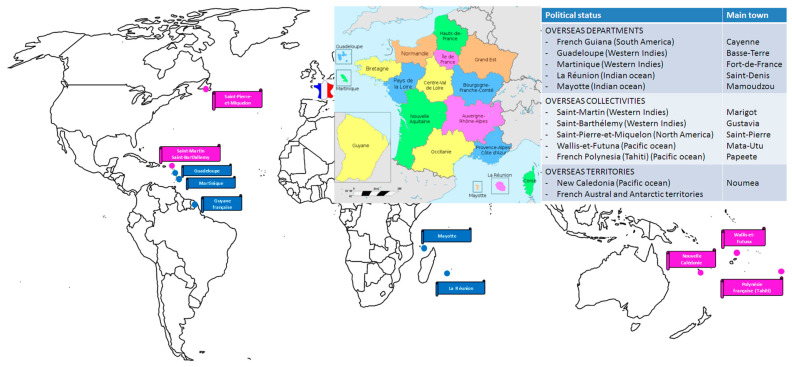
Location of French overseas territories and administrative regions. Overseas departments are shown in blue. Overseas collectivities and territories are shown in magenta (French Austral and Antarctic Territories not shown). The embedded map shows the 18 French administrative regions.

**Figure 2 viruses-15-01783-f002:**
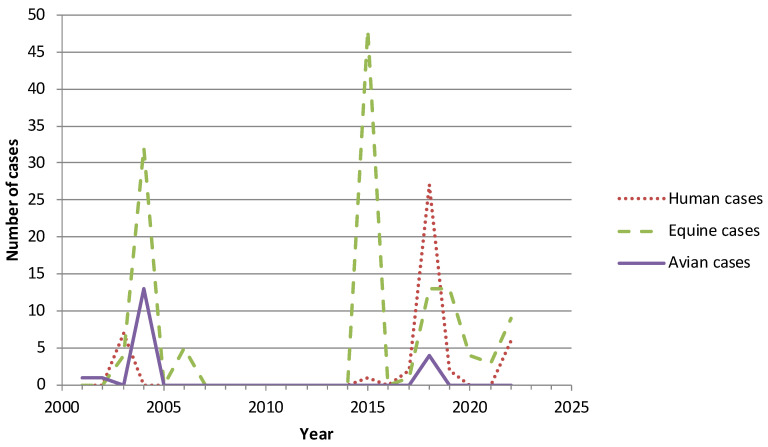
Number of annual cases of West Nile virus infection identified in humans, horses, and birds in mainland France, 2001–2022 (source: Santé publique France).

**Figure 3 viruses-15-01783-f003:**
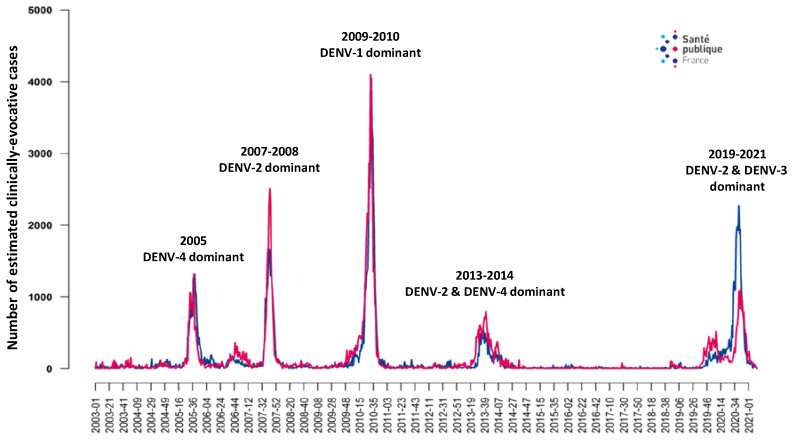
Endemic and epidemic circulation of dengue virus on Martinique (blue line) and Guadeloupe (red line) islands, 2003–2021. The more prevalent serotype(s) is (are) shown for each outbreak. Source: Santé publique France.

**Figure 4 viruses-15-01783-f004:**
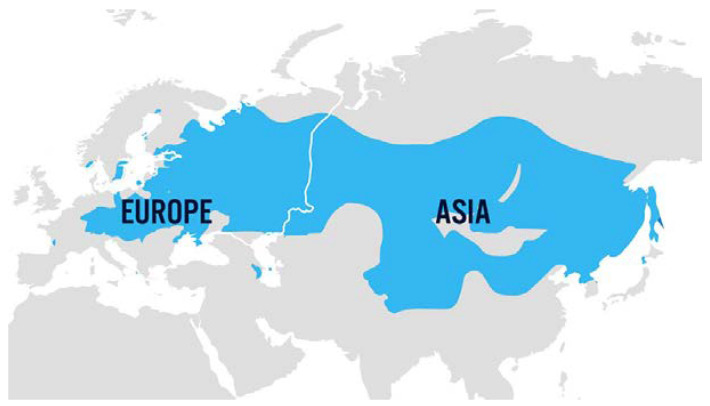
Worldwide distribution of tick-borne encephalitis virus (TBEV). Source: Centers for Disease Control and Prevention, 7 March 2022 (available at https://www.cdc.gov/tick-borne-encephalitis/geographic-distribution/index.html).

**Table 1 viruses-15-01783-t001:** Examples of arboviruses presenting a potential or demonstrated risk of inter-human transmission by blood or grafts. For viruses with more than one publication documenting inter-human transmission through blood or graft, the references are not included in the table.

Virological Data *Family*/*Genus*	Vectors	Main Vertebrate Hosts	Geographic Distribution	Incubation (Days)	Percent of Asymptomatic Forms	Main Clinical Symptoms	Current Vaccine Prophylaxis	Documented Transmission by Blood or Grafts (Reference)
** *Flaviviridae* ** **/** ** *Flavivirus* ** ** (single-stranded RNA, enveloped)**								
**Dengue virus** **(DENV, serotypes 1 to 4)**	Mosquitoes (*Aedes aegypti* and *Aedes albopictus*)	Humans Non-human primates	World (mainly intertropical regions)	2–14	50–75	Fever, hemorrhagic dengue, shock	Yes	Yes (numerous cases, see text and tables)
**Japanese encephalitis virus** **(JEV)**	Mosquitoes (genus *Culex*)	Pigs Water birds	South-East Asia and Western Pacific	4–15	>99	Fever, headache, encephalitis	Yes	Yes [2]
**Powassan virus** **(POWN)**	Ticks (genera *Ixodes* and *Dermacentor*)	Rodents, deer, and other wild mammals	North America Russia	7–30	Frequent	Fever, headache, vomiting, weakness, (meningo-)encephalitis	No	Yes [3]
**Saint-Louis encephalitis virus** **(SLEV)**	Mosquitoes (genus *Culex*)	Birds	Americas	4–21	>99	Fever, encephalitis	No	Yes [4]
**Tick-borne encephalitis virus** **(TBEV)**	Ticks (genera *Ixodes* and *Dermacentor*)	Rodents Deer	Europe, Asia	2–28	80	Fever, encephalitis	Yes	Yes (see text)
**Usutu virus** **(USUV)**	Mosquitoes (genus *Culex* but also *Aedes*)	Birds	Africa, Israel, Europe	3–6	Frequent	Fever, rash, (meningo-)encephalitis	No	No
**West Nile virus** **(WNV)**	Mosquitoes (genus *Culex* but also *Aedes albopictus*)	Birds	Asia, Africa, Europe, Americas	2–14	80	Fever, encephalitis	No	Yes (numerous cases, see text and tables)
**Yellow fever virus** **(YFV)**	Mosquitoes (*Aedes aegypti*)	Humans Non-human primates	Africa Central and South America	3–6	55	Fever, jaundice Hemorrhagic fever, shock	Yes	Yes with YFV vaccine [5]
**Zika virus** **(ZIKV)**	Different mosquitoes (genera *Aedes*, *Anopheles*, *Mansonia*)	Humans Non-human primates	Africa, Oceania, India, South-East Asia, Western Indies, Central and South America, Europe	3–12	30–80	Fever, rash, conjunctivitis, arthralgia, myalgia, Guillain–Barré syndrome, microcephaly	No	Yes (see text)
** *Reoviridae* ** **/** ** *Coltivirus* ** ** (double-stranded RNA, naked)**								
**Colorado tick fever virus** **(CTFV)**	Ticks (*Dermacentor andersoni*)	Humans	Western USA, Canada	3–6	Low	Fever, encephalitis	No	Yes [6]
** *Togaviridae* ** **/** ** *Alphavirus* ** ** (single-stranded RNA, enveloped)**								
**Chikungunya virus** **(CHIKV)**	Mosquitoes (*Aedes aegypti* and *Aedes albopictus*)	Humans Non-human primates	Africa, Asia, Europe, Indian and Pacific oceans, Western Indies, Americas	1–12	15	Fever, fatigue, arthralgia	No	Yes [7]
**Ross river virus** **(RRV)**	Different mosquitoes (genera *Culex*, *Aedes*, *Anopheles*, *Mansonia*)	Kangaroos and wallabies	Oceania, South Pacific	5–15	50–75	Fever, rash, arthralgia	No	Yes [8]

**Table 2 viruses-15-01783-t002:** Reported cases of transmission of West Nile virus following solid organ transplantation (adapted from [19]).

Year/Country (Reference)	Donor		Recipient(s)						
	Infection Route	Serum Testing	Organ	OSPT ^1^	Serum Testing	CSF Testing	Treatment	Symptoms	Outcome
2002/USA [15,16]	Blood transfusion	NAT ^2^ +	Kidney Kidney Heart Liver	13 17 8 7	IgM+ IgM− NAT+ IgM+	IgM+ IgM− IgM+ NoNAt tested	None None None None	Neuroinvasive disease Neuroinvasive disease Neuroinvasive diseaseFever	Survived Died ^3^ Survived Survived
2005/USA [20]	Probable mosquito bite	NAT– IgM+ IgG+	Liver Lung Kidney Kidney	13 17 AS ^4^ AS	IgM+ IgM+; IgG+ NAT+; IgM−; IgG+ NAT−; IgM−; IgG−	NAT+; IgM+ NAT+; IgM+ Not tested Not tested	Immunotherapy Immunotherapy Immunotherapy Immunotherapy	Neuroinvasive disease Neuroinvasive disease Asymptomatic Not infected	Coma Coma Survived Survived
2008/USA [21]	Blood transfusion	NAT−; IgM+	Heart	8	IgM+	IgM+	Supportive care	Neuroinvasive disease	Survived
2009/USA [22]	Probable mosquito bite	NAT+; IgM−	Liver	15	NAT−; IgM+; IgG-	IgM+	Immunotherapy	Neuroinvasive disease	Survived
2009/USA [23]	Mosquito bite	NAT+; IgM+; IgG equivocal	Kidney Kidney Liver	NA ^5^ AS AS	NA NA NA	NA NA NA	NA NA NA	Neuroinvasive disease AsymptomaticA symptomatic	Survived Survived Survived
2009/Italy [24]	Mosquito bite	NAT+	Liver	AS	NAT+; IgM+	Not tested	Immunotherapy	Asymptomatic	Survived
2010/USA [23]	Mosquito bite	NAT+; IgM−; IgG+	Kidney Kidney Liver	8 AS AS	IgM+; IgG+ NAT+; IgM+; IgG+ NAT−; IgM−; IgG+	NAT−; IgM+ Not tested Not tested	Supportive care None None	Neuroinvasive disease Asymptomatic Asymptomatic	Died Survived Survived
2011/Italy [25]	Mosquito bite	NAT−; IgM+; IgG+	Kidney Kidney Liver Lung Heart	10 10 AS AS AS	NAT+; IgM+; IgG+ NAT+; IgM+; IgG+ NAT−; IgM+; IgG+ NAT+; IgM+; IgG+ NAT−; IgM−; IgG−	NAT+; IgM+; IgG+ NAT+; IgM+; IgG+ Not tested Not tested Not tested	Immunotherapy None None None None	Neuroinvasive disease Neuroinvasive disease Asymptomatic Asymptomatic Not infected	Coma Survived Survived Survived Survived
2011/USA [26]	Increased WNV activity in the donor region	NAT−; IgM+; IgG+ (NAT+ in lymph nodes and spleen)	Kidney Kidney Lungs Liver	10 17 20 AS ^7^	NAT+ NAT+; IgM+ NAT+ NAT−; IgM−; IgG+	NAT+; IgM− NAT+; IgM− NAT+; IgM+ NAT+; IgM−	Immunotherapy + IFNα2b ^6^ Immunotherapy + IFNα2b Immunotherapy + IFNα2b Immunotherapy + oral ribavirin	Neuroinvasive disease Neuroinvasive disease Neuroinvasive disease No sign of WNV infection	Died Survived Survived Survived

^1^ OSPT: onset of symptoms post transplantation. ^2^ NAT: nucleic acid testing. ^3^ The patient’s brain was NAT-positive at autopsy. ^4^ AS: asymptomatic. ^5^ NA: not available. ^6^ IFN α2b: Interferon α2b. ^7^ The patient developed symptoms at day 18 post transplantation, but not in connection with an WNV infection.

**Table 3 viruses-15-01783-t003:** Published cases of possible transplantation-associated infection by dengue virus (DENV). NS1. Ag: NS1 antigen of DENV.

Country (Reference)	Period	Donor			Recipient(s)					Accountability
		Sex/Age/Status	Viral Diagnosis	Clinical Picture	Graft	Sex/Age	Viral Diagnosis	Clinical Picture	Evolution	
Puerto Rico [43]	1994	NA ^1^/NA/Alive	Not tested	Fever 2 days after gift	Bone marrow	NA/6	Not tested	Not reported	Deceased	Low
Singapore [44]	Not reported	F/NA/Alive	Not tested	Fever	Kidney	M/23	PCR+ (DENV-1)	Hemorrhagic dengue	Survived	Intermediate
Thailand [45]	Not reported	F/NA/Alive	Positive serology without details	Fever one month before gift	Kidney	F/16	NS1 Ag+ PCR+ (DENV-1) Culture+	Hemorrhagic dengue	Survived	Low
India [46]	Not reported	M/19/Alive	NS1 Ag+ PCR+ (type?)	Not reported	Liver	M/38	NS1Ag+ PCR+ (type?)	Fever + hepatitis	Survived	Strong
Germany [47]	2013	F/24/Alive	IgM weakly + IgG weakly + NS1 Ag+ PCR+ (DENV-1)	Fever one day before gift, 8 days after return from Sri Lanka	Bone marrow	M/51	IgM+/IgG+ NS1 Ag+ PCR+ (DENV-1)	Leukemia worsening Enterocolitis Hepatic veno-occlusive disease	Death not dependent of dengue	Very strong (same sequence)
India [48]	Not reported	M/29/Alive	NS1 Ag+	Fever 3 days after gift	Liver	M/40	NS1 Ag+	Fever	Survived	Strong
Columbia [49]	2007	M/40/Deceased	IgM+/IgG+	Fever	Liver	M/53	IgM+/IgG− PCR+ (DENV-3)	Fever + transient encephalopathy + hepatitis	Survived	Strong
					Heart	M/41	IgM−/IgG− PCR+ (DENV-3)	Hemorrhagic dengue + shock	Survived	Strong
	2010	M/32/Deceased	IgM−/IgG+ NS1+	Asymptomatic	Kidney	F/31	IgM−/IgG− NS1 Ag+ PCR+ (DENV-4)	Hemorrhagic dengue	Survived	Strong
					Kidney	F/48	IgM+/IgG− NS1 Ag−, PCR-	Fever	Survived	Strong
India [50]	2016	58/M/Alive	IgM+/IgG+ NS1 Ag− PCR+ (DENV-1)	Fever 6 days after gift Encephalopathy	Liver	M/58	IgM−/IgG− NS1 Ag+ PCR+ (DENV-1)	Fever + encephalopathy + liver and kidney failure	Deceased	Very strong (same sequence)
La Reunion, France [51]	2020	62/M/Deceased	IgM+/IgG+ PCR−	Asymptomatic	Kidney Kidney	M/58 M/61	PCR+ (DENV-1) IgM+/IgG+ PCR+ (DENV-1) IgM+/IgG+	Pancytopenia + hepatic cytolysis + hemorrhagic shockIntraabdominal collection operated infected by *S. epidermidis* Thrombopenia + hepatic cytolysis	Survived Survived	Strong Strong

^1^ NA: Not available.cornea graft [52]. The donor, a 76-year-old Indian, was deceased from hemorrhagic fever; although the donation was recused, the corneoscleral tissue was cultivated, leading to the identification by NAT of the presence of DENV RNA that was further typed 3.

## Data Availability

The material used for this review is available in the References section.

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
