# Peer review of "Arboviral Risk Associated with Solid Organ and Hematopoietic Stem Cell Grafts: The Prophylactic Answers Proposed by the French High Council of Public Health in a National Context"

_viruses, 2023, doi:10.3390/v15091783_

Round 1
Reviewer 1 Report
This manuscript is well written and succinctly summarizes the history and risk of arboviral transmisssion through blood products and tissue transplantation. No critique.
Author Response
PLease find enclosed my answers to the three reviewers.
Many thanks for your help.

Reviewer 2 Report
The scholarly article titled "Arboviral Risk Associated with Solid Organ and Hematopoietic Stem Cell Grafts: Prophylactic Approaches Proposed by the French High Council of Public Health in a National Context," authored by Bruno Pozzetto and colleagues, demonstrates a commendable level of composition and addresses an intriguing subject within the realm of public health.
Within the manuscript, there exist only minor issues that warrant correction.
Table 1 presents an assortment of viral features, drawing from the review's discussions. It is advisable to ensure proper referencing of the various articles that contributed to the compilation of these features.
Regrettably, Figure 1 suffers from blurriness, thereby muddling the clarity of the information presented. Consequently, this figure fails to contribute effectively to the review's contextual narrative.
Likewise, Table 2 necessitates proper referencing, as it draws upon articles used by the authors to encapsulate the data contained therein succinctly.
Attention is warranted towards lines 416-424, specifically in relation to the font employed in this section. A discrepancy is evident when juxtaposed with the rest of the manuscript, prompting the need for consistency in font usage.
In conclusion, while the manuscript by Bruno Pozzetto and colleagues is well-executed and brings forth a thought-provoking public health topic, these outlined issues merit revision for an enhanced overall presentation.
none
Author Response

(The authors gave the same response as above.)

Reviewer 3 Report
Overall, Pozetto et al. present an interesting and important manuscript. Pozetto et al. review the main arboviruses occurring in France and French overseas territories and regions. They discuss the main risks for human products (blood, organs and hematopoietic stem cells) and present the various guidelines. They conclude that a "One Health" approach is imperative and international collaboration to standardize recommendations is needed.
My major comment is about the way this review has been conducted. I understand the authors opted for a narrative review. However, I wonder whether a systematic review following the PRISMA 2020 guidelines would not have been more advanced in methods to identify, select and critique the various studies. Please consider this option.
The rest of my comments are minor suggestions.
Please find my comments in the attached pdf.

The English could be slightly improved. I have provided feedback as much as I could.
Author Response

(The authors gave the same response as above.)
